# Implication of the Gut Microbiome and Microbial-Derived Metabolites in Immune-Related Adverse Events: Emergence of Novel Biomarkers for Cancer Immunotherapy

**DOI:** 10.3390/ijms24032769

**Published:** 2023-02-01

**Authors:** David Dora, Syeda Mahak Zahra Bokhari, Kenan Aloss, Peter Takacs, Juliane Zsuzsanna Desnoix, György Szklenárik, Patrick Deniz Hurley, Zoltan Lohinai

**Affiliations:** 1Department of Anatomy, Histology, and Embryology, Semmelweis University, Tuzolto St. 58, 1094 Budapest, Hungary; 2Translational Medicine Institute, Semmelweis University, 1094 Budapest, Hungary; 3Department of Thoracic Surgery, Guy’s Hospital, London SE1 9RT, UK; 4National Korányi Institute of Pulmonology, Pihenő út 1-3, 1121 Budapest, Hungary

**Keywords:** irAEs, ICI, anti-PD1 immunotherapy, gut microbiome, metabolome

## Abstract

Immune checkpoint inhibitors (ICIs) have changed how we think about tumor management. Combinations of anti-programmed death ligand-1 (PD-L1) immunotherapy have become the standard of care in many advanced-stage cancers, including as a first-line therapy. Aside from improved anti-tumor immunity, the mechanism of action of immune checkpoint inhibitors (ICIs) exposes a new toxicity profile known as immune-related adverse effects (irAEs). This novel toxicity can damage any organ, but the skin, digestive and endocrine systems are the most frequently afflicted. Most ICI-attributed toxicity symptoms are mild, but some are severe and necessitate multidisciplinary side effect management. Obtaining knowledge on the various forms of immune-related toxicities and swiftly changing treatment techniques to lower the probability of experiencing severe irAEs has become a priority in oncological care. In recent years, there has been a growing understanding of an intriguing link between the gut microbiome and ICI outcomes. Multiple studies have demonstrated a connection between microbial metagenomic and metatranscriptomic patterns and ICI efficacy in malignant melanoma, lung and colorectal cancer. The immunomodulatory effect of the gut microbiome can have a real effect on the biological background of irAEs as well. Furthermore, specific microbial signatures and metabolites might be associated with the onset and severity of toxicity symptoms. By identifying these biological factors, novel biomarkers can be used in clinical practice to predict and manage potential irAEs. This comprehensive review aims to summarize the clinical aspects and biological background of ICI-related irAEs and their potential association with the gut microbiome and metabolome. We aim to explore the current state of knowledge on the most important and reliable irAE-related biomarkers of microbial origin and discuss the intriguing connection between ICI efficacy and toxicity.

## 1. Introduction

Immune checkpoint inhibitors (ICI) and their combination with conventional chemotherapy and/or radiotherapy are currently the standard of care in cancer, and the number of patients receiving ICI as a first-line therapy is rising. Currently, the CTLA-4 inhibitor ipilimumab; PD-1 inhibitors pembrolizumab, nivolumab, and cemiplimab; and PD-L1 inhibitors atezolizumab, avelumab, and durvalumab are approved by the US Food and Drug Administration (FDA) for the treatment of metastatic melanoma, non-small cell lung cancer (NSCLC), Hodgkin’s lymphoma, head and neck squamous cell carcinoma, urothelial carcinoma, renal cell carcinoma and hepatocellular carcinoma [1]. PD-L1 inhibitors are also approved for the treatment of triple-negative breast cancer and Merkel cell tumors [2].

ICIs have now been in use for a decade and, in addition to their positive effects, there is growing evidence of their side effects. The efficacy of ICIs, including their toxicity, have been demonstrated in murine models to have a significant link to the gut microbiome [3]. All aspects of the gut, including the microbiome, micro-organism composition and metabolic products, induce effects on host immune defenses [4]; however, in healthy individuals, tolerance and antigenic response are generally in balance. In recent years, numerous independent research groups discovered an intriguing link between the gut flora and ICI efficacy in a clinical setting [5,6]. Multiple investigations and an impartial meta-analysis in malignant melanoma demonstrated a connection between microbial metagenomic and metatranscriptomic markers and ICI efficacy [7,8,9]. Despite the fact, that causality and the biological background of such a connection are still disputed and under scrutiny, multiple studies have showed at least partial or indirect evidence regarding such mechanisms. The most likely supported hypothesis is the “molecular mimicry” phenomenon, where epitopes produced by gut microbial species as part of their native gene expression programs resemble tumor neoantigens, prompting “autoreactive” T cells and powerful anti-tumor immunity [10,11]

Similar to other anti-cancer therapies, ICI comes with its own set of adverse effects; about 70–90% of patients receiving ICI show immune-related adverse events (irAEs) [12,13,14,15]. irAEs are commonly attributed to the inability of the immune system to distinguish between tumor and self-antigens, hence the cross-reactivity [16]. This dysregulation of the immune system’s self-tolerance leads to an over activation of innate and adaptive immunity that translates to cytokine mediated inflammation, autoreactive B and T cells and autoantibodies [17,18]. irAEs can be observed in almost all organs of the human body, more commonly the skin, thyroid, lungs, colon, liver and pituitary are affected; however, in rare cases even the nervous and cardiac systems can be damaged, with the possibility of a fatal outcome [18,19]. The mechanisms underlying these toxicities are diverse and differ in each organ; in contrast, irAEs can have similar manifestations in the affected organ regardless of the cancer type [20,21,22].

In this review, we aim to summarize the epidemiology and clinical features of irAEs non-comprehensively, then discuss the immune-related cellular and molecular events predating irAE occurrence during cancer immunotherapy. Additionally, we aim to summarize the current state of knowledge on the connection between irAEs and the gut microbiome, including the microbial metabolome and metabolic pathways. Due to the fact that efficacy and toxicity of ICI treatments are often derived from similar immunological mechanisms, we also cover the current state of knowledge on the connection between microbial signatures and ICI efficacy and the phenomenon of efficacy–toxicity coupling.

## 2. Adverse Events and Their Clinical Management in Anti-Cancer Immunotherapy

irAEs are classified by the American International Cancer Institute according to the Common Terminology Criteria for Adverse Events (CTCAE), which is divided into mild (1), moderate (2), severe (3), life-threatening (4) and fatal (5) categories [23]. PD-1 and PD-L1 inhibitors are generally better tolerated than CTLA-4 inhibitors and grade 3 and 4 irAEs are also more common with CTLA-4 inhibitors than with PD-1 inhibitors (31% vs. 10%). There is also a difference in the organ specificity of adverse events, for example, colitis, hypophysitis and skin rashes occur more frequently with CTLA-4 inhibitors, while pneumonia, hyper- or hypo-thyroidism, joint pain and vitiligo are more common with PD-1 inhibitors [23]. Generally, at least one negative event is detected in two-thirds of anti-PD-1/PD-L1 immunotherapies. One-seventh of patients experience a high grade irAE (3–5) and 0.45% of patients die, most commonly due to pneumonitis [24]. The most common symptoms of treatment-associated irAEs (at any grade) were fatigue, pruritus, diarrhea, hyperthyroidism, hypothyroidism and vitiligo; while the most common high grade irAE symptoms were fatigue, liver enzyme elevation (AST and ALT), pneumonitis and diarrhea [13,15,25]. Figure 1 demonstrates the most common and severe irAEs organized by organ systems.

### 2.1. Dermatological irAEs

Dermatological toxicity is one of the most common irAEs among all organ systems, occurring in 25–30% of treated patients. Clinically, immune-related cutaneous toxic effects are mainly skin rashes, pruritus and vitiligo [26,27] the latter being a side effect specific to the treatment of melanoma. It has been shown that the treatment of skin tumors is most frequently associated with the highest incidence of dermatological side effects (7.3 times more frequent than other tumors, [28]). Interestingly, the presence of cutaneous irAEs showed positive correlation with treatment response and long-term relapse-free survival, according to a recent meta-analysis [27]. Topical agents or low-dose systemic steroids are often sufficient to treat skin-related irAEs [29].

### 2.2. Gastrointestinal irAEs

Pooled meta-analysis data suggest that diarrhea (13% vs. 33%) and colitis (1.4% vs. 9.1%) are less frequent with PD-1/PD-L1 inhibitors compared to CTLA-4 therapy, but symptoms persist longer (2 months vs. 1.4 months) [30,31,32]. Unlike CTLA-4 therapy, the severity of diarrhea as an adverse effect of PD-1/L1-blockade is not dose-dependent [33]. The most common symptom of colitis is diarrhea, which occurs in almost all patients, followed by abdominal pain (25%), loss of appetite (19%), hematochezia (12.5%) and mucoid stools (10%). GI pathology includes inflammatory infiltrates, villi shortening and crypt or mucosal fragility. GI irAEs present clinically as described above—colitis, ileitis, decreased transit time, diarrhea and/or blood in stool (ICI-therapy associated colitis/ileitis, CIC) [34,35]. The risk of colitis is increased by taking NSAIDs and is decreased in the case of vitamin D supplementation [30]. Autoimmune diseases increase the risk of developing CIC, and CIC occurrence in underlying IBD patients receiving anti-CTLA4 therapy was reported at 30% [36,37]. In contrast, IBD patients better tolerate anti-PD1/PD-L1 ICI [31].

Low grade GI toxicity is usually managed based on clinical symptoms. The gold standard for the diagnosis of CIC is biopsies, but to exclude a more severe situation, such as toxic megacolon or perforation, CT scans are required. An infection can be excluded by obtaining a stool sample for culture [31]. Treatment of colitis in grade 1 is supportive, with antimotility agents (loperamide), hydration and appropriate diet to reduce symptoms, which can be continued during therapy. In the case of grade 2, systemic steroid treatment should commence, after the exclusion of an infection. In grades 3 and 4, immediate discontinuation of ICI therapy is required, hospital admission for monitoring is recommended, along with IV administration of 1–2 mg/kg/day methylprednisolone with electrolyte and fluid therapy. If symptoms do not improve after 2–4 days, biological therapy (infliximab and vedolizumab) should be administered. While therapy should not be restarted at grade 4, opinions differ at grade 3, but if symptoms can be managed and reduced to grade 1 or below, PD-1/PD-L1 therapy may be restarted [33]. In grades 3–4 cases, 1–1.5% of patients develop colon perforation, which is treated with emergency colon resection surgery [30].

The most common hepatic irAE is hepatitis, with around 1–2% occurrence in PD-1/PD-L1 monotherapy, but this can be as high as 20% in a combination of ICI therapy with ipilimumab [37]. Blood testing of liver enzymes is essential before therapy initiation, as is monitoring ICI throughout the therapy and before each dose [38]. Treatment in the first two grades is based on continuous monitoring of enzyme levels, and at grade 2, initiation of oral steroid therapy. In grades 3 and 4, glucocorticoids is given in a high dose for the first two days, followed by 1–2 mg/kg/day of oral prednisolone until enzyme levels settle [32]. To date, there are some published case reports of pancreas involvement [39,40] but research has shown that PD-1/PD-L1 inhibitors do not increase the risk of pancreatitis [41].

### 2.3. Respiratory irAEs

Most fatal adverse events are caused by pneumonitis (35%) [42] compared to other irAEs, but the severe, life-threatening grade is found in less than 2% of cases [38]. The real-word incidence of immune-related pneumonitis in ICI-treated patients is between 2.49% and 13.2% [43,44,45]. The most common symptoms are a persistent, unproductive cough, dyspnea, fever and chest pain; however, one-third of patients remain asymptomatic [46]. Risk factors include elderly age, smoking, male gender, previous lung disease (emphysema, COPD or asthma), previous chest irradiation and combination with other therapies [42]. Diagnosis is based on a CT scan and clinical symptoms. Bronchoscopy or BAL helps with differential diagnosis to rule out an infection [47].

Treatment of ICI-related grade 1 pneumonitis includes the temporary suspension of the drug until the normalization of CT findings. Grade 2 treatments include 1 mg/kg/day of oral prednisolone and in grades 3–4, complete abandonment of ICI-therapy is required with 2–4 mg/kg/day IV methylprednisolone for 4–6 weeks and initiation of antibiotic therapy to rule out infection [47].

### 2.4. Endocrine irAEs

Hypothyroidism, hyperthyroidism and hyperglycemia are the most common endocrine irAEs. Grade 3 or higher side effects are mostly due to hypoglycemia, adrenal insufficiency and Type 1 DM [25]. Approximately 40–50% of patients experience some change in thyroid function during therapy [46,48,49]. Subclinical hyperthyroidism occurs in 18%, overt hyperthyroidism in 12%, subclinical hypothyroidism in 5% and overt hypothyroidism in 3% of all cases [46]. A total of 7.7% of patients who develop persistent hypothyroidism require levothyroxine administration [48]. Onset of overt thyrotoxicosis occurred after a median of 5 weeks of the receipt of first cycle. Combination ICI therapy, female gender and younger age were strongly associated with the development of overt thyrotoxicosis [46]. Intriguingly, even in real-world practice, ICI treatment-induced thyroid dysfunction was associated with better outcomes [50,51,52].

Immunotherapeutic monoclonal antibodies can damage healthy tissues directly, for example, anti-CTLA4 treatment directly damages the pituitary [53]. The incidence of hypophysitis is 1–18% in metastatic melanoma patients treated with anti-CTLA4, and 0.5–1.5% for PD-1 inhibitors [54,55]. However, the incidence can rise to up to 13% in the case of combination therapy [56,57]. There have also been cases of PD-1/PD-L1 therapy-induced diabetes (0.2%). Although uncommon, 81% of cases occurred with diabetic ketoacidosis and, unlike thyroiditis, all cases developed fully even with prednisolone usage [58]. Generally, endocrine side effects are always treated with hormone replacement and non-steroidal therapy after the acute phase [59]. 

### 2.5. Cardiovascular irAEs

Cardiovascular side effects of ICI therapy pose the greatest challenge to physicians and include myocarditis, pericardial disease, supraventricular arrhythmia and vasculitis. Anti-PD1 antibodies can directly damage heart tissue, causing concomitant myositis and rhabdomyolysis and robust infiltration of macrophages and T cells [60,61,62,63]. A systematic review reported that the most frequent underlying biological mechanisms were the recruitment of CD4+ and CD8+ T cells, autoantibody-mediated cardiotoxicity and substantial inflammatory cytokine release [64]. Supraventricular arrhythmias are almost always associated with cardiac irAEs and are therefore thought to be secondary events [65]. In a review of several meta-analyses, myocarditis was found to be among the less common irAEs at about 1% prevalence, but has an extremely high mortality (27–46%) compared to others [66]. Risk factors include elderly age (70–80 years), male sex, ethnicity and pre-existing autoimmune or cardiovascular disease [67,68]. Symptoms usually occur in the form of palpitations, dyspnea and left ventricular pump dysfunction, as well as chest pain, hypotension, lower limb oedema or heart block, but less specific symptoms such as fatigue, malaise, ptosis, diplopia, paresis, nausea and vomiting can also dominate. In severe cases, hemodynamic instability, cardiogenic shock and sudden death may happen [66,67].

Close cardiological monitoring and temporary drug suspension are considered in asymptomatic cardiac involvement due to asymptomatic arrhythmias or structural abnormalities of the myocardial wall. If symptoms are present, discontinuation of therapy is recommended, then, depending on improvement, restarting treatment can be considered [38]. If myocarditis is suspected, immediate hospital admission is necessary and high dose of corticosteroid should be administered as the first choice. In severe or life-threatening cases, pulsatile methylprednisolone at 1 g/day for 3–5 days is recommended and, in refractory cases, mycophenolate mofetil or tacrolimus should be used. If the patient has symptoms of chronic heart failure, β-blockers and ACE inhibitor/ATR2 antagonists are administered as treatment [66].

### 2.6. Musculoskeletal irAEs

Musculoskeletal side effects caused by PD-1/PD-L1 inhibitors occur in between 2 and 12% of cases, with around one in 15 patients requiring rheumatological treatment [69]. The most common presentations are arthralgia, inflammatory arthritis, sicca syndrome, myositis, vasculitis and polymyalgia rheumatica. Treatment with NSAIDs is recommended for grade 1 arthritis, a low dose (10–20 mg/day) and a high dose (0.5–1 mg/kg/day) of prednisone is recommended for grade 2 and grade 3 arthritis, respectively, with the temporary discontinuation of therapy. Disease-modifying antirheumatic drugs (DMARDs) are recommended as steroid-sparing agents in steroid-resistant cases and for steroid reduction [70].

### 2.7. Neurological irAEs-Peripheral Nervous System

Neurological irAEs include involvement of the peripheral nervous system (PNS) and the neuromuscular unit and central nervous system afflictions. The most common manifestations of ICI-related peripheral neurological syndromes are myositis, myasthenia gravis (MG) and Guillain–Barré Syndrome (GBS). Immune-related myositis is a rare irAE and has been reported mainly in melanoma patients [71]. Its clinical symptoms differ significantly from those of idiopathic and paraneoplastic inflammatory myopathies such as dermatomyositis (DM) and polymyositis (PM) [72,73]. There have also been reports of dyspnea, dysarthria and dysphonia. Despite this, the clinical pattern is very consistent, with myalgia being the most prevalent and early symptom, even in the absence of creatine kinase (CK) elevation [73].

ICI-related MG is a well-known neurological irAE that has been extensively documented in case reports and case series. This condition rarely arises without associated myositis, according to growing data [74]. Individuals suspected of having an irMG should receive a thorough evaluation that includes a CK, an electromyogram and, if possible, a muscular MRI and a biopsy in order to discover associated myositis and alter treatment and follow-up. The presence of irMG and ir-myositis at the same time raises the likelihood of a myasthenic crisis, which may necessitate ventilator support and hospitalization in an intensive care unit. Bulbar symptoms, dysarthria, dysphagia and dyspnea occur in 50% of the cases [75].

In terms of treatment regarding irMG and irMyositis, pyridostigmine is commonly used, but rarely as a singular therapy (3–9%). Often, immune modulatory treatment is required, either with steroid alone (27–45%) or in combination with immunoglobulin infusion and/or plasma exchange (50–63%) [76,77].

GBS is an uncommon complication among irAEs, occurring in around 0.1–0.3% of patients treated with ICIs [78,79,80]. However, according to an earlier meta-analysis, this peripheral neuropathy was more common (up to 3% for anti-PD-L1 drugs and 7% for anti-PD-1) [81]. In addition, a systemic evaluation of 86 patients treated with ipilimumab or pembrolizumab found that 23% had some form of demyelinating polyradiculoneuropathy [82]. Acute classical GBS begins with subtle paresthesia, followed by leg weakness, then arm, face and oropharyngeal weakness as it spreads proximally (ascending paralysis). Pain is prevalent, manifesting as bilateral sciatica or pain in the large muscles of the upper legs [83].

### 2.8. Neurological irAEs-Central Nervous System

The most common manifestations of CNS irAEs are encephalitis, meningitis [84,85] transverse myelitis [86,87,88] multiple sclerosis (MS)-like demyelination syndromes [89], vasculitis [90] and cranial neuropathies (CNDs) [91], however, myelitis, MS, vasculitis and CNDs are extremely rare irAEs and evidence of their presence is mainly based on case reports and case series.

The most common and severe irAEs with CNS involvement are aseptic meningitis and encephalitis. In cases of acute or subacute onset of headache, altered mental status, psychiatric symptoms, speech impairments, seizures or neurological deficits with/without fever, immune-related encephalitis (irEncephalitis) should be suspected and be differentiated from infectious encephalitis, CNS metastasis or metabolic encephalopathy [92]. According to Larkin and colleagues, irEncephalitis occurred in 0.16% patients treated with ICI and accounted for 0.44% of all irAEs [84]. A total of 82% of encephalitis cases occurred alone without the manifestation of other irAEs [85]. The incidence of encephalitis is higher following anti-PD-1/PD-L1 treatment compared to after anti-CTLA-4 therapy and is more associated with combination therapy compared to monotherapy [93]. Given the high fatality rate of CNS irAEs (encephalitis, 6.3–12.8% and meningitis, 7.4–8.3 [93,94] a prompt recognition of suspected patients is required and treatment with immune-modulating agents should start without delay. Diagnostic tests to confirm neurological irAEs include lumbar puncture, cytology and tests for infectious diseases with neuroimaging.

irAEs often occur in a non-specific form that is difficult to diagnose and may onset several months after therapy [95]. In addition, the side effects that are more likely to cause death vary from drug to drug. In the case of CTLA-4 inhibitors colitis is more likely; in the case of PD-1/PDL1 inhibitors pneumonitis, hepatitis and neurotoxicity are more likely; and in the case of combination therapy, colitis and myocarditis are more likely [19]. Therefore, a complex multidisciplinary approach is essential in the clinical management of ICI treatment to prevent severe and life-threatening irAEs, including early monitoring or even prophylactic supportive care. Still, we lack the inclusion of high-sensitivity, high-specificity biomarkers in everyday clinical practice that can predict the onset of irAEs in a timely manner, which can endanger therapy or the health of patients.

## 3. The Biological Basis of irAEs in Anti-Cancer Immunotherapies

T cells are the most important players in the immune defense and play an equally important role in irAEs. Mechanisms by which T cells induce toxic effects are diverse and can be organ dependent but most commonly include an imbalance in the ratio of CD8+ and CD4+ cells [96]. Apart from activating tumor cell-specific T cells, systemically delivered ICI can activate previously inactivated self-recognizing T cells, which leads to autoimmune reactions [97]. Furthermore, an increase in cytokine and chemokine production can lead to inflammation, which can has adverse effect in that it promotes [98]. In many cases, ICI reduces regulatory T (Treg) cell proliferation and activity, abolishing T cell response regulation, thereby increasing CD8+ T cells in the tumor peripheral area, which promotes autoimmune inflammation [99,100]. Treg cells further reduce the cytotoxic activity of T cells by hindering the function of antigen presenting cells (APCs) and cause T cell exhaustion by an enhanced expression of anti-inflammatory IL-6 [101]. ICI causes diversification of the CD8+T cell repertoire [102] as an increase in highly proliferative and cytotoxic phenotype of CD8+T cells was observed in the colon of patients treated with anti-PD1 ICI. Similarly, an increase in CD8+T cell diversity was observed in patients receiving anti-CTLA4 therapy. This diverse repertoire of immunological actions could interestingly both benefit and harm the patient, as the combination of toxicity and benefits results from a complex and divergent pathway which converges at the diversification of T cells [54,96,103].

CD4+T cells are classified to different T helper types, namely Th1, Th2, Th17 and follicular T cells (Tfh). Both the IFN-γ-producing Th1 cells and IL-17-producing Th17 cells promote autoimmune reactions, leading to colitis, nephritis, liver damage and skin complications [16]. Furthermore, patients receiving ICI often have a much higher Th17/Th1 cell ratio [104]. This increase in Th17 cells leads to increased inflammation and autoimmune adversities in patients [105]. Melanoma patients receiving anti-PD1 therapy demonstrated an increase in Th-1-mediated IL-6 production which led to severe colitis [106].

ICIs directly affect B cell function by reducing the number of circulating B cells while increasing production of the CD21-low phenotype. These CD21-low B cells have high IFN-γ production that enhances immune response and might be responsible for B cell exhaustion [107]. B cells produce auto-antibodies in response to ICI against self-reactive T and B cells [108,109] which commonly damage the thyroid and islet cells of the pancreas leading to thyroiditis, hypothyroidism and diabetes, respectively [50,110].

Other cells of the immune system also partake in irAEs directly or indirectly. Activated neutrophils promote T and B cell-mediated response and lead to distant inflammation and organ damage with their long lasting effects [111,112]. Likewise, eosinophils promote irAEs by producing inflammation-promoting IL-17, which can elicit an inflammatory response; however, the effects mediated by eosinophils do not seriously affect patient survival [113]. NK cells are classically tumor suppressive; however, their functionality is altered in response to ICI in many cancers [114]. Either the NK cells become hyperactive and produce pro-inflammatory cytokines, or they modulate the immune functions of dendritic cells, T and B cells and the epithelium, resulting in an exaggerated inflammatory response and damage to hepatocytes [115].

ICI treatment can induce macrophage activation and accumulation in patients with anti-PD1 treatment, which is translated to muscle weakness, atrophy and myopathy [116]. In response to ICI, exhausted T cells release IFN-γ, which recruits monocyte-derived macrophages. These macrophages acquire cytotoxic abilities and are reported to damage the pancreas leading to diabetes [117]. Furthermore, with an active T cell response and Treg reduction, the infiltration of type 2 macrophages promotes further inflammation and leads to progressive organ damage [118]. In patients receiving ICI, activated monocytes were reported to promote liver inflammation, coagulation and fibrinolysis pathways and hyperactivation of the innate immune system, leading to dermatitis [94,119,120].

Although involvement of these immune cells in promoting irAEs is clinically evident, the mechanistic basis and potential cross-talks need further elucidation. Moreover, the role of other factors such as genetics, epigenetics, environment and predisposition to an autoimmune disease cannot be neglected. Equivalently, gut microbiota are another contributing factor that can alter immune function by cross-reactivity, as an elevated level of microbial antibodies has been detected in ICI-receiving patients [121] which might result in the competition with self or tumor antigens or the recruitment and activation of immune cells [102]. Species of *Bacteroides* and *Burkholderiales* provide protection against irAEs; *Bacteroides fragilis*, for example, reduces irAEs by promoting Treg cell development and production of anti-inflammatory IL-10 [122]. On the contrary, high *Firmicutes* phylum can promote colitis in melanoma patients by supposedly sequestering Treg cells [123].

The mechanisms by which the immune system reacts to and regulates the effects of ICI are diverse and complex. Several pre-clinical and clinical trials are currently underway to identify specific antibodies, cytokines and/or pathways to circumvent these mild to severe irAEs and improve the usability of ICI treatments worldwide [124]. Figure 2 shows the immune modulatory role of ICI treatment in different immune cells.

## 4. Microbial Signatures Associated with irAEs

It has been suggested that the composition of the microbiome might play a role in the success of immunotherapy, and by modulating ICI, might influence survival and the development of side effects. There are a plethora of species residing in healthy guts, including *Bacteroidetes*, *Clostridiales* and *Firmicutes* phyla, all of which have crucial functions in the host [125]. Certain species of bacteria have been identified which aid ICI response and decrease ICI toxicity, including *Akkermansia*, *Bifidobacterium*, *Faecalibacterium*, *Lactobacillus* and *Ruminococcaceae spp*.

The latest studies suggest that ICIs in vivo are not effective in the absence of a normal intestinal microbiome. Tan et al. described that mice undergoing antibiotic therapy fail to respond to anti-CTLA-4 therapy, but also appear to have subclinical irAEs. However, it was also described that antibody treatment itself may alter or destroy the intestinal microbiome [125]. It is known that symbiotic microbes have the ability to induce colitis via an interleukin-1β (IL-1β) mechanism [126]. Murine models treated with (IL-1β) receptor antagonists had a significantly reduced level of inflammation [127]. However, it was also found that pre-treatment with antibiotics had a similar effect, even after ICI administration.

As described by Davar and Zarour across 11 studies investigating similar pathologies, it is difficult to draw a good conclusion regarding different microbial signals which instigated a clinical response or irAEs [128] which can be explained by the highly diverse and multifaceted system of microbiome–immune system interactions that is influenced by many factors, including diet, medication use, geography and ethnicity [128]. At the highest interpreted phylogenetic level, Dubin et al., reported that resistance to development of irAEs was related to the *Bacteroidetes* phylum, whereas increased frequency of irAEs was associated with *Firmicutes* [13,129]. Notwithstanding these known microbe that influence irAEs, Hayase and Jenq described a phenomenon in which antibiotic administration prior to ICI therapy does result in a reduced response to therapy; however, it does not impact the frequency nor the severity of irAEs [129]. Matson and colleagues have identified eight species that are present in those who show lower rates of toxicity to ICI therapy, the so-called “good bacteria”. These include *Bifidobacterium adolescentis*, *Bifidobacterium longum*, *Collinsella aerofaciens*, *Enterococcus faecium*, *Klebsiella pneumoniae*, *Lactobacillus sp.*, *Parabacteroides merdae* and *Veillonella parvula* [8]. Absence of *Akkermansia muciniphila* has also been found in mice that are nonresponsive to treatment, and thus have increased rates of irAEs. The restoration of *A. muciniphila* using oral supplementation restored the PD-1 blockade against epithelial tumors [130].

Emerging research has demonstrated that the presence or overrepresentation of some bacterial strains or species may increase irAEs, specifically colitis. *Bacteroides* species, as well as members of *Enterobacteriaceae, Klebsiella pneumoniae* and *Proteus mirabilis*, can promote or induce colitis during ICI therapy [131]. Two bacteria have been identified that increase irAEs in melanoma anti-PD-1 treatments, namely *Lachnospiraceae* spp. and *Streptococcus* spp. [132,133]. Patients treated with anti-PD-1 (ipilimumab) who had an increased representation of *Faecalibacterium* and *Firmicutes* also exhibited an increased ICI sensitivity and efficacy; however, they had an increased risk of developing colitis or other ICI-induced toxicities [132,134]. This effect is known as the efficacy–toxicity coupling effect [19,132,134,135] which may complicate the management of ICI-treated cancer patients and have a significant impact on outcomes [35].

Several studies have demonstrated fecal microbiota transplantation (FMT) as a way to augment the efficacy of ICI treatments, and thus decrease irAEs [7,8,130]. FMT treatment is the transplantation of the entire microbiota of an individual to another individual. This can be done by oral lyophilized pills or directly by using colonoscopy or gastroscopy. FMT was generally reserved for treatment of resistant *Clostridium difficile* infection [136]. though ICI has demonstrated some usefulness in patients where anti-PD-1 therapy was no longer effective or in cases with severe CIC. Two separate studies have shown that in previously treatment-resistant or non-responding patients, FMT treatment restored ICI therapy efficacy and decreased irAEs in melanoma [137,138].

Fungi also have a significant role in the development of and changes in the gut tolerance and antigenic reaction balance. Fungal infections or adverse reactions caused by overcolonization are generally rare [139]. To date, no concrete fungi species have been directly implicated in the development of irAEs. However, ICI treatment is generally administered with corticosteroids, which increases the likelihood of a fungal, as well as bacterial, overgrowth [140]. ICI therapy and its relationship with gut microbiota and toxicity is becoming a more important field when treating patients for a variety of different cancers. With the demonstration of the importance of FMT in responsiveness to ICI, it is clear that the gut microbiome can significantly affect treatment in patients, as well as decrease irAEs. Table 1(a,b) shows microbial taxa implicated in decreased and in increased irAEs, respectively.

## 5. Microbial Metabolites and Metabolic Pathways Associated with irAEs

Apart from commensal microbial communities, metabolites of gut microbiota might also influence ICI efficacy and toxicity. Short-chain fatty acids (SCFAs) (e.g., butyrate, propionate, acetate and valeric acid), indole derivatives, tryptophan derivatives, polyamines and B vitamins play critical roles in inflammation and immunity—locally and systemically—comprising the maintenance of epithelial barrier function [149,150]. Furthermore, some of these microbial metabolites might be involved in the development of diseases such as cardiovascular diseases and diabetes [150,151]. SCFAs can inhibit the development of irAEs by interacting with immune cells and suppressing the production of pro-inflammatory cytokines [152]. For instance, the low production of butyrate, the main energy source for colonocytes [153] was shown to contribute to PD-1/PD-L1 inhibitor-induced cardiotoxicity in a B16F10 melanoma mouse model. Interestingly, butyrate supplementation and mice recolonization with *Prevotella loescheii*, a SCFA-producing strain, reduced the PD-1/PD-L1-related myocardial apoptosis by the activation of PPARα. PPARα activation inhibited mediators of PD-1/PD-L1 inhibitor-related cardiotoxicity, including the prevention of NF-κB-mediated M1-like polarization of colonic macrophages and downregulation of pro-inflammatory factors such TNF-α and IL-1 [154]. In accordance with this finding, butyrate production was found to be reduced in melanoma patients who experienced severe irAEs. However, serum butyrate levels were not affected compared to patients with mild irAEs [155]. Moreover, SCFAs have been reported to be histone deacetylase (HDAC)-inhibitors [156] which have been found to reduce the severity of colitis in mice [157]. Lactate is another metabolite that exhibited anti-inflammatory activity [100,158,159] and immunomodulatory effects [160]. The rectal administration of lactate reduced the early production of IL-6 in a murine model of colitis [158]. In accordance with this finding, lactate supplementation reduced the inflammation severity and restored the function of the intestinal epithelium by improving the expression of the tight junction proteins ZO-1 and mucin (MUC-2) throughout the colonic mucosa in colitis mice. Furthermore, lactate supplementation elevated the content of SCFAs in the feces of the colitis mice [159].

Similar to SCFAs, polyamines such as spermine are expected to confer protection against the development of irAEs. Zhang et al. revealed that spermine inhibited the production of pro-inflammatory cytokines in vitro. In addition, the administration of spermine exhibited anti-inflammatory effects in mice [161] while others have shown that indole-3-carboxaldehyde (3-IAld), a microbial tryptophan metabolite, can protect against ICI-induced colitis due to ICI’s role in preserving the integrity of the intestinal barrier [162]. The administration of 3-IAld reduced the inflammatory cytokine production and improved the expression of zonula occludens-1 (ZO-1), a tight junction protein, in a mouse model of CIC. Moreover, 3-IAld supplementation increased the fecal levels of butyric acid, valeric acid and propionic acid [162]. However, 3-IAld was shown to be defective in the serum of mice with active colitis but not in patients with active colitis [163]. Trimethylamine N-oxide (TMAO) is a microbial metabolite that exhibited synergism with anti-PD-1 antibodies by promoting CD8+ T cell-mediated anti-tumor immunity in a triple-negative breast cancer (TNBC) mouse model [164]. In addition, the high plasma levels of TMAO were associated with a better response to anti-PD-1 in TNBC patients [164]. However, the elevated plasma TMAO is linked with an increased risk of cardiac diseases [165,166] thus, the correlation between ICI-induced cardiotoxicity and circulating TAMO requires thorough investigation. Similar to SCFAs and polyamines, the secreted bacterial fragments can also interact with the immune response and thereby influence the development of irAEs. Round et al. showed that polysaccharide A promoted the production of the anti-inflammatory interleukin-10 (IL-10) [167] which is a regulator of the mucosal immune response. Furthermore, IL-10 deficiency has been linked with the development of colitis [168].

Most of the studies have assessed the plasma and fecal levels of microbial metabolites as potential biomarkers of response to ICI. High fecal levels of acetic acid, propionic acid, butyric acid, valeric acid and plasma isovaleric acid were associated with longer progression-free survival (PFS) in patients with solid tumors treated with PD-1 inhibitors [169]. Similarly, non-small cell lung cancer (NSCLC) patients with high fecal propionate had a longer PFS after anti-PD-1 treatment. However, the metabolomic profiling of patients with early progression revealed low fecal levels of SCFAs, especially propionic acid, nicotinic acid and lysine. On the other hand, 2-pentanone, tridecane and p-cresol were abundant in the feces of those patients [170] and in addition, p-cresol exhibited genotoxicity on colon epithelial cells in vitro [171]. In contrast to the previous findings [169,170] both low baseline butyrate and low baseline propionate (in serum) were correlated with longer PFS in metastatic melanoma patients treated with anti-CTLA-4 antibodies [172].

In addition to concrete metabolites, several metabolic pathways have shown association with the clinical outcome of ICI. These pathways were identified using metagenomic sequencing of stool samples obtained before or during ICI treatment. Dubin et al. investigated the pathways that could be involved in the resistance to anti-CTLA-4- induced colitis in metastatic melanoma patients by comparing colitis-free patients and patients with colitis. This prospective study revealed higher abundance of the pathways involved in the polyamine transport system (spermidine/putrescine) and the biosynthesis of B vitamins, including thiamine (B1), riboflavin (B2), pantothenate (B5) and biotin (vitamin B7) in colitis-free patients [146]. Similarly, pathways of vitamin B synthesis (thiamine and biotin), butyrate production, purine degradation and amino acid synthesis (tryptophan and methionine) were reduced in ICI non responder metastatic melanoma patients who had severe irAEs [155]. These findings indicate that B vitamins can be used as biomarkers to predict the development of colitis in patients undergoing ICI. This suggestion is supported by the finding that the low serum levels of B vitamins (B9 and B12) were linked to IBD [173]. However, the pathways of B vitamin biosynthesis (pantothenate, pyridoxal 5-phospate and flavin) were shown to be associated with shorter PFS in melanoma patients receiving ICI (anti-PD-1 and/or anti-CTLA-4) [174].

Apart from irAEs, metabolic pathways of gut microbiota might predict the response to ICI. Frankel et al., showed an enrichment in fatty acid synthesis pathways in metastatic melanoma patients who responded to all kinds of ICI. In addition, an increase in inositol phosphate metabolism was observed among responders to a combination of ipilimumab plus nivolumab [175]. Similarly, the biosynthesis of fatty acids, acarbose and polyketide sugar units was abundant in hepatobiliary cancer patients who responded to anti-PD-1 treatment, and amino acid biosynthesis pathways (arginine) were enriched in the non-responder group [176]. Zheng et al. showed that pathways of carbohydrate metabolism and methanogenesis were predominant in hepatocellular carcinoma patients responding to anti-PD-1 ICI [177]. In contrast, methanogenesis was found to be abundant in metastatic melanoma patients who did not respond to ICI [178]. Table 2 summarizes the association of irAEs and ICI efficacy with microbial metabolites and pathways.

## 6. The Efficacy–Toxicity Coupling Effect: A Crucial Challenge for Future Clinical Practice

An intriguing question in this field is whether irAE occurrence is connected to better ICI response and long-term benefits or not. In melanoma, a study has found that younger patients with ≥G3 irAEs show significantly increased overall survival (OS) compared to others, but the same conclusion could not be unequivocally made in older adults [179]. A pooled cohort study described a significantly better overall response rate (ORR), but no significant difference in progression-free survival (PFS) in metastatic melanoma patients with any grade of irAEs [55]. In lung cancer, patients who experienced irAEs exhibited an improved PFS and OS compared to those who did not [180,181,182]. In renal cell carcinoma, an improved OS in patients with irAEs was reported by two independent studies [135,183] and similar conclusions were made in patient cohorts with colorectal and gastric cancer [184,185] and in the case of head and neck malignancies [186]. A systematic review screening 52 articles and 9156 patients with various cancers revealed significantly better ORRs and PFS in patients with irAEs, especially in NSCLC and melanoma in the case of skin- and endocrine organ-related toxicities [187].

Furthermore, multiple studies have suggested an association between the timing of irAE occurrence and ICI efficacy, where an early irAE onset predicted improved objective response and PFS for multiple cancers [187,188,189]. Despite the commonplace and routine use of corticosteroids in irAE management, there is growing evidence that steroid use can hamper ICI efficacy according to a melanoma and NSCLC study [190,191]. The growing evidence of an intimate link between autoimmunity and anti-tumor effects makes the early and proper management of ICI-related irAEs a vital task for clinicians and warrants the development of innovative therapies to robustly suppress dangerous irAEs in a way to maintain an appropriate immune response. Endeavors to uncouple the autoimmune effect of ICIs from their efficacy are ongoing, including a clinical trial in metastatic melanoma patients where ipilimumab is used in combination with GM-CSF [192]. Another high-potential possibility is the targeting of the pleiotropic cytokine IL-6, which proved to be successful in an experimental autoimmune encephalomyelitis model, and a clinical trial is also on the way [193].

## 7. The Prospect of the Microbiome in Future Cancer Therapies

An estimated 3.8 × 10^13^ bacteria are found in the microbiome of the whole human body and its most diverse and abundant system is the commensal flora of the gut [194]. The gut microbiome has been a source of discussion in the recent scientific literature, with an emerging focus on its role in anti-cancer therapies. Functions include, but are not limited to, influences on the efficacy of immunotherapy, association with irAEs and a potential target for new anti-cancer treatments.

### 7.1. FMT

Building on these findings, recent preclinical research has shown that microbiome modulation via administration and subsequent gut colonization of commensal bacteria, including *Bacteroidales*, *Bifidobacterium* and *Akkermansia muciniphila*, can enable and increase the efficacy of blockade therapies [130,195,196,197]. There are multiple ongoing trials that are studying the role of FMT in patients that are not responsive to ICI therapy in gastrointestinal cancers (NCT04130763, Phase 1), prostate cancer (NCT04116775, Phase 2) and melanoma (NCT03353402, Phase 1 and NCT03341143, Phase 2). The proposition of these trials is to investigate if response to immunotherapy improves once a favorable gut microbiome is formed after FMT.

### 7.2. Nanotechnologies

Another promising approach, nanotechnology, is gaining traction in microbiota modulation, where specific bacteria are selected to regulate the proliferation or metabolism of the commensal microbiome. While still in its early stages, seminal work already exists that demonstrates its enormous promise. Based on previous demonstrations, we believe that the technology’s existing toolbox can be used to design novel approaches to navigate the microbial microenvironment by targeting the gut [198] targeting specific microbes [199] delivering to inflammation [200] penetrating and diffusing across the mucus [201] and transporting across the epithelium into systemic circulation [202]. Beyond the gut microbiome, in the tumor microenvironment (TME), nanotechnologies have the ability to target and act on migrating tumor-associated bacteria (TAB) and their metabolites which remain in distal tumors [203,204]. Specific aspects of non-gut-associated TMEs provide options for targeting TAB using nanotechnologies, such as delivering payloads to hypoxic tumor regions [205,206]. Nanotechnologies, which have previously been used to facilitate immunotherapy by interacting with specific cell populations in the tumor [207] have the potential to interfere with persistent TAB or TAB–cell interactions.

Different strategical approaches include the addition and deletion of beneficial or cancer-causing bacterial species by modulating the existing commensal population [208]. Promotion of bacterial species secreting the short chain fatty acid butyrate [209,210] is one of these strategies. Antibiotics might seem to be the first choice for the targeted killing of selected bacteria and modulate the microbiome; however, the use of broad-spectrum antibiotics can result in antimicrobial resistance and erratic changes in the commensal microbiome that have been demonstrated to reduce immunotherapy efficacy [211] and to cause dysbiosis or inflammation [212] both of which have been linked to tumor formation [213,214]. Nanotechnologies have been used for a decade for the loading and delivery of antibiotics to kill bacteria, including cancer-causing bacteria [215,216,217,218,219]. Nanotechnologies to intervene in the human microbiome are ahead in the same challenges faced by current cancer nanotechnologies, including targeting efficiency, biodistribution, adverse effects, scale-up and delivery challenges caused by tumor heterogeneity [220].

### 7.3. CRISPR/Cas9

Another interesting approach is to deplete specific commensal bacteria, whose selective depletion is not possible with antibiotics that exert erratic effects to the whole microbiome. CRISPR-Cas9 technology enables the creation of antimicrobials that delete a designed spectrum of microbes and allow non-targeted bacteria to [221]. Recent studies on mice have already demonstrated that the elimination of *Escherichia coli*, *Enterococci* or *Citrobacter rodentium* in the gut microbiome was achievable with CRISPR-driven genetic engineering [222,223,224]. Targeting specific bacteria with minimal disturbance to the commensal microbiome will act as a further target in the optimization of future anticancer therapy.

### 7.4. Viruses and Bacteriophages

Bacteriophages are *Duplodnaviria* viruses that target, infect and replicate within bacteria [225]. The therapeutical utilization of viruses and bacteriophages to improve ICI efficacy or decrease irAEs has already been proposed, mainly from a theoretical perspective [226,227,228]. Delivery of personal tumor antigens by fabricated phage systems [66] or development of human phage-derived anti-PD1 antibodies [229] are some of the recent innovative endeavors to create an efficient adjuvant therapeutic platform for ICI. A recent study found that the tail length tape measure protein (TMP1) in a bacteriophage targeting E. hirae reduced MCA205 cell-line sarcoma growth, but only when E. hirae 13144 was co-administered with cyclophosphamide or PD1 inhibitors in murine models [230,231]. An ongoing clinical trial proposes that CAN-2409 viral immunotherapy could potentially induce T cells to further infiltrate tumor cells, and consequently upregulate PD-L1, enhancing ICI therapy (NCT04495153).

The approach of phage therapy to modulate gut microbiota has been gaining traction in recent years, mainly to eliminate antibiotic-resistant strains and treat dangerous or recurrent GI infections [232,233,234,235]. Bacteriophages have also been introduced as potential immunomodulatory agents, boosting cellular immunity and antigen recognition to confer adjuvanticity [236,237]. Despite the rising number of efforts in the field, no direct studies on gut microbiome manipulation in the context of ICI-related irAEs and efficacy have been performed so far. Due to the fact that a technological platform is already available, phage-mediated microbiome engineering [238,239,240] is a promising therapeutic approach and should be given high priority in the future.

ICI has become an extremely successful modality for the treatment of multiple cancer types, including hematological and metastatic malignancies. A more detailed understanding of the function of specific bacteria will provide critical insight into the ever-growing role of the gut microbiome. With its emerging effects on immunotherapy becoming clear, the modulation of the gut microbiome provides us with the promise of optimizing anti-cancer therapies and enhancing patient care.

## Figures and Tables

**Figure 1 ijms-24-02769-f001:**
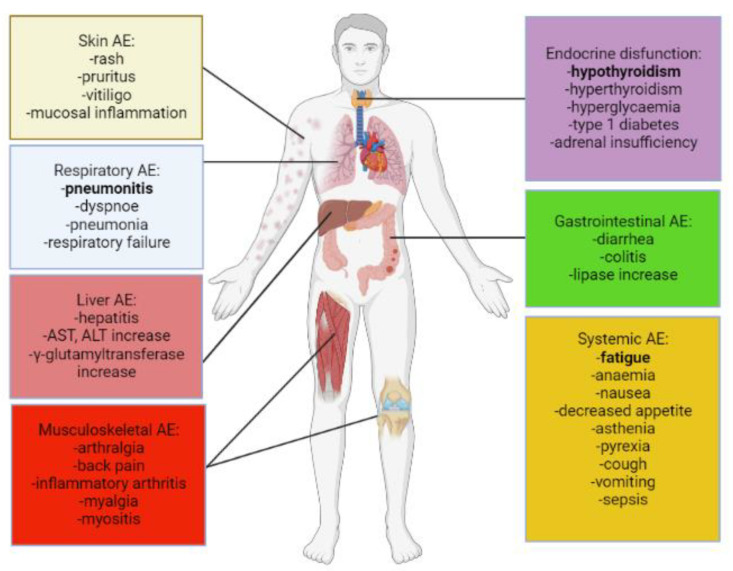
Most common and significant irAEs of PD-1/PD-L1 inhibitors. Figure shows the most common side effects organized by organ system. The most common treatment-associated systemic irAE at any grade was fatigue. Among all irAEs, diarrhea and thyroid dysfunction (hypo- or hyper-thyreoidism) occurred most frequently. Pneumonitis is was most common high grade symptom and most frequent cause of irAE-related death.

**Figure 2 ijms-24-02769-f002:**
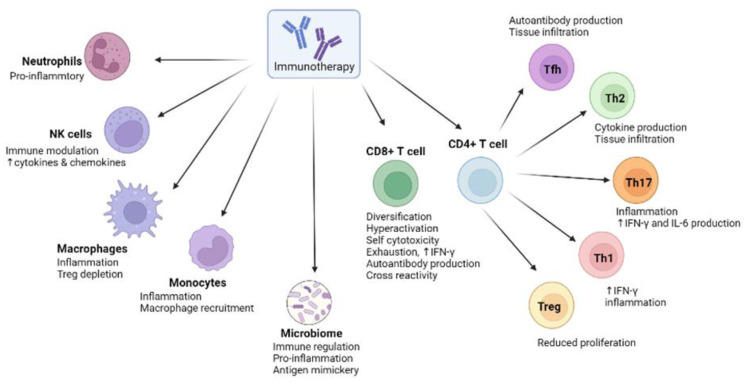
Immune modulatory role of immune cells in response to ICI blockade. Immunotherapy mediates functional alteration of immune cells and the microbiome, leading to hyperactivation, reduced proliferation, self-cytotoxicity, autoantibody production and cross-reactivity. This complex modulation of the immune system facilitates a pro-inflammatory and autoreactive immune environment which translates to immunotherapy-related adverse events.

**Table 1 ijms-24-02769-t001:** Microbial taxonomic units associated with irAEs.

(a) Summary of the Microbial Taxonomic Units Associated with Decreased irAEs					
Bacteria Decreasing ICI Toxicity	Associated Disease/Intervention	Reported Adverse Reactions	Microbiome Analysis Method	Patient Number	Reference
*Bacteroidetes (fragilis*, *Barnesiellaceae*, *ikenellaceae*, *Bacteroidaceae*)	Kidney/Lung/Ovary/Stomach Cancer	Diarrhea, Bloody Stool	16s rRNA	30	[141]
*Akkermansia muciniphilia*	Metastatic Melanoma	None	16s rRNA	42	[8]
	Metastatic Non-small Cell Lung Cancer	Pneumonia, Pneumonitis, Bronchopleural fistula	16s V4 rRNA	53	[142]
*Bifidobacterium*	Lung Cancer/PD-L1/Anti-PD-L1	Colitis, Myositis Rash, Thrombocytopenia, Pneumonitis	16s rRNA	13	[143]
	Metastatic Melanoma	None	16s rRNA	42	[8]
*Faecalibacterium*	Metastatic Melanoma/Anti-CTLA-4	Colitis	16s rRNA	26	[134]
*Lactobacillus*	Metastatic Melanoma	None	16s rRNA	42	[7,8]
*Ruminococcaceae*	Solid Tumors/Nivolumab + Ipilimumab	Pneumonia, Pneumonitis, Bronchopleural fistula	16s V4 rRNA	53	[142]
	Malignant Melanoma	None	16s rRNA	112	[7]
*Burkholderia cepacia*	Malignant Melanoma/Lung Cancer/GI tumors	Colitis	N/A	None	[144]
*Dorea formicigenerans*	Malignant Melanoma/Lung Cancer/GI tumors	Colitis	N/A	None	[144]
*Caloramator coolhaasii*	Cutaneous Melanoma, Lung Cancer	Colitis, Ileal Damage	16s rRNA; Shotgun sequenceing	77	[126]
*Anaerococcus vaginalis ATCC 51170*	Cutaneous Melanoma, Lung Cancer	Colitis, Ileal Damage	16s rRNA; Shotgun sequenceing	77	[126]
*Anaerotignum lactifermentans*	Cutaneous Melanoma, Lung Cancer	Colitis, Ileal Damage	16s rRNA; Shotgun sequenceing	77	[126]
*Geosprorobacter* unclassified	Cutaneous Melanoma, Lung Cancer	Colitis, Ileal Damage	16s rRNA; Shotgun sequenceing	77	[126]
*Proteobacteria Desulfovibrio*	Lung Cancer/PD-L1/Anti-PD-L1	Colitis, Myositis Rash, Thrombocytopenia, pneumonitis	16s rRNA	13	[143]
*Acinetobacter* spp.	Solid Tumors (Non-Small Cell Lung Cancer (NSCLC), Small Cell Lung Cancer (SCLC), Hepatocellular Carcinoma (HCC), and Renal Cell Carcinoma (RCC) / N/A)	Colitis, Mysoitis, Neurologic, ICI Efficacy	N/A	433 (across multiple studies)	[6]
*Collinsella aerofaciens*	Metastatic Melanoma	None	16s rRNA	42	[8]
*Enterococcus faecium*	Metastatic Melanoma	None	16s rRNA	42	[8]
*Dialister* unclassified	Non-Small Cell Lung Cancer	Colitis	16s rRNA	1010	[145]
**(b) Summary of the Microbial Taxonomic Units Associated with Increased irAEs**					
**Bacteria Increasing ICI Toxicity**	**Associated Disease/Intervention**	**Reported Adverse Reactions**	**Microbiome Analysis** **Method**	**Patient Number**	**Reference**
*Faecalibacterium prausnitzii* and *Gemmiger formicilis*	Metastatic Melanoma	Colitis	Shotgun Sequencing; 16 s rRNA	34	[146]
*Klebsiella pneumoniae*	Metastatic Melanoma	N/A	16s rRNA	42	[8]
*Proteus Mirabilis*	Solid Tumors (Cutaneous Melanoma/Non-small Cell Lung Cancer)/Anti-PD1/PDL1/Anti-CTLA4	Rash, Respiratory, Genitourinary, Bacteremia	Routine Clinical Bacterium Testing	327	[147]
*Lachnospiraceae*, *Streptococcus* spp.	Melanoma	Adrenal, Arthritis, Dermatologic, Colitis, Hepatitis, Neurologic, Thyroid, Pneumonitis	16s rRNA	57	[133]
*Faecalibacterium*	Solid Tumors (Cutaneous Melanoma/Non-small cell lung cancer)/Anti-PD1/PDL1/Anti-CTLA4	Rash, Respiratory, Genitourinary, Bacteremia	Routine Clinical Microbial Test	327	[147]
*Bacteroides Intestinalis*	Melanoma/Anti-CTLA-4	Colitis	16s rRNA	26	[134]
*Intestinibacter bartlettii*	Cutaneous Melanoma, Lung Cancer	Colitis, Ileal Damage	16s rRNA	77	[127]
*Firmicutes*	Melanoma	Colitis	16s rRNA	26	[134,148]
*Parabacteroides ditasonis*	Melanoma	Colitis	N/A	N/A	[5]
*Faecalibacterium prausnitzii and Gemmiger formicilis*	Melanoma	Colitis	16s rRNA	26	[134,148]

**Table 2 ijms-24-02769-t002:** Summary of the microbial taxonomic units associated with increased irAEs.

Metabolite/Metabolic Pathway	Disease/Model	Sample	Results	Reference
*Butyrate*	Mouse model	Stool	Low butyrate contributed to ICI-induced cardiotoxicity	[154]
*Butyrogenesis*	Metastatic melanoma	Stool, serum	Low butyrogenesis in patients with severe ICI- adverse effects Serum butyrate did not change	[155]
*3-IAld*	Mouse model	Stool, serum	3-IAld supplementation to mice with ICI-induced colitis protected from intestinal damage	[162]
*TMAO*	TNBC	Plasma	High plasma TMAO associated with longer PFS	[164]
*Acetic acid* *Propionic acid* *Butyric acid* *Valeric acid* *Isovaleric acid*	Solid tumors	Stool, Plasma	High SCFAs in feces (acetic acid, propionic acid, butyric acid and valeric acid) and plasma (isovaleric acid) associated with longer PFS	[169]
*Propionate* *Nicotinic acid* *Lysine*	Lung cancer	Stool	Low levels in patients with early progression	[170]
*2-Pentanone* *Tridecane* *p-cresol*	Lung cancer	Stool	High levels in patients with early progression	[170]
*Butyrate* *Propionate*	Metastatic melanoma	Serum	Low levels associated with longer PFS	[172]
*Polyamine transport system* *B vitamins biosynthesis*	Metastatic melanoma	Stool	Abundant in colitis free patients	[146]
*B vitamins biosynthesis* *Purine degradation* *Amino acid synthesis*	Metastatic melanoma	Stool	Reduced in ICI-non responders who experienced severe adverse effects	[155]
*B vitamins biosynthesis*	Metastatic melanoma	Stool	Associated with shorter PFS	[174]
*Fatty acid synthesis* *Inositol phosphate metabolism*	Metastatic melanoma	Stool	Abundant in responders	[175]
*Fatty acid synthesis* *Acarbose synthesis* *Polyketide sugar unit synthesis*	Hepatobiliary cancer	Stool	Abundant in responders	[176]
*Amino acid synthesis (arginine)*	Hepatobiliary cancer	Stool	Abundant in non-responders	[176]
*Carbohydrate metabolism* *Methanogenesis*	Hepatocellular carcinoma	Stool	Abundant in responders	[177]
*Methanogenesis*	Metastatic melanoma	Stool	Abundant in non-responders	[178]

## Data Availability

Not applicable.

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
