# Peer review of "Implication of the Gut Microbiome and Microbial-Derived Metabolites in Immune-Related Adverse Events: Emergence of Novel Biomarkers for Cancer Immunotherapy"

_ijms, 2023, doi:10.3390/ijms24032769_

Round 1
Reviewer 1 Report
Overall, I think this review by David Dora et al discusses an important area. I only have minor suggestions.
Figures provided in this review should be accompanied by figure legends, to get a better understanding of what points does the portrayed figure intend to emphasize.
There are several sentence construction or language issues in Section 7, for eg, not sure if the information of 3.8x1013 bacteria inhabiting the body is correct, is 13 supposed to be superscript, and found in the human body means which part of the human body?. Line 615 should be addition "and" deletion instead of "end". line 618 should be might, not migh. Please check for similar factual or sentence errors.
Author Response
We thank the expert reviewer for the positive and constructive review of our article.
We marked all changes with a green highlight in the revised version of the manuscript.
Figures provided in this review should be accompanied by figure legends, to get a better understanding of what points does the portrayed figure intend to emphasize.
We thank the Reviewer for pointing this out. Accordingly, we added a more detailed figure legend for both Figure 1 and 2.
There are several sentence construction or language issues in Section 7, for eg, not sure if the information of 3.8x1013 bacteria inhabiting the body is correct, is 13 supposed to be superscript, and found in the human body means which part of the human body?. Line 615 should be addition "and" deletion instead of "end". line 618 should be might, not migh. Please check for similar factual or sentence errors.
We thank the Reviewer for spotting these mistakes. Accordingly, we corrected these inconsistencies in chapter 7, plus we performed one more round of spelling check to improve the English language of the manuscript.
Reviewer 2 Report
This is a well-written paper by David Dora and colleagues summarizing the role of the Gut Microbiome and Microbial-derived Metabolites in Immune-related Adverse Events in patients undergoing anti-cancer therapy treated with Immune-Checkpoint Inhibitors.
Author collected and reviewed recent literature regarding the connection between efficacy and toxicity of anti-cancer therapy with ICI, its immune-related adverse effects (irAEs) and the gut microbiome conditions, including the microbial metabolome and metabolic pathways. These data were linked with Review is organized as follows: 1. Introduction; 2. Adverse events and their clinical management in anti-cancer Immunotherapy; 3. The biological basis of irAEs in anti-cancer Immunotherapies; 4. Microbial signatures associated with irAEs; 5. Microbial metabolites and metabolic pathways associated with irAEs; 6. The efficacy-toxicity coupling effect: a crucial challenge for future clinical practice; 7. The prospect of the microbiome in future cancer therapies.
Author extensively discussed latest literature data. Paper has 231 references, which are relevant to article's subject.
Manuscript is illustrated with 2 figures explaining „ Most common and significant irAEs of PD-1/PD-L1 inhibitors.” and „ Immune modulatory role of immune cells in response to ICI-blockade”. Illustrations are very informative, helpful and provide great data presentation. Along figures two tables are presented: Table 1. Summary of the microbial taxonomic units associated with decreased irAEs / Summary of the microbial taxonomic units associated with increased irAEs; Table 2. Summary of the microbial taxonomic units associated with increased irAEs.
This is an interesting review-study, and clinically valuable, especially for those researchers and clinicians who use ICIs modalities and meet irAEs during anti-cancer therapies. This manuscript provides comprehensive background information on this issue.
Minor comments:
Comment 1. Section 5. Microbial metabolites and metabolic pathways associated with irAEs.
I would extend this section with short paragraph on the role of bacterial metabolite: L-lactate (D-lactate), which is an obvious product of commensal microbiota metabolism. Lactate can operate as agonist of HCAR1 receptor as well as HDAC inhibitor. Lactate strongly affects cells within TME, including immune cells Treg, macrophages and others; See examplery papers PMID: 27119568, PMID: 33123172, PMID: 34899735.
Comment 2. I am curious whether there is an option to use bacteriophages, which are specific bacterial viruses to modulate microbiome composition in a way that would enhance ICIs effectivity or diminish irAEs.
Comment 3. Text line 241: Please explain the abreviation DMARDs
Text line 583: superscript of 13 in 3.8 x 1013
Text line 615: e>a; the addition and deletion of beneficial….
Taken together, this manuscript by David Dora and colleagues represents a worthwhile contribution to the cancer research. I recommend the manuscript for further publication process.
Author Response
We thank the expert reviewer for their constructive criticism and excellent suggestions to improve our manuscript.
We marked all changes with a green highlight in the revised version of the manuscript.
Comment 1. Section 5. Microbial metabolites and metabolic pathways associated with irAEs.
I would extend this section with short paragraph on the role of bacterial metabolite: L-lactate (D-lactate), which is an obvious product of commensal microbiota metabolism. Lactate can operate as agonist of HCAR1 receptor as well as HDAC inhibitor. Lactate strongly affects cells within TME, including immune cells Treg, macrophages and others; See examplery papers PMID: 27119568, PMID: 33123172, PMID: 34899735.
We thank the Reviewer to bring up these important studies and so we included a paragraph about the role of the metabolite lactate in the topic. We also included the suggested and relevant citations. See changes in section 5, end of first paragraph.
Comment 2. I am curious whether there is an option to use bacteriophages, which are specific bacterial viruses to modulate microbiome composition in a way that would enhance ICIs effectivity or diminish irAEs.
We thank the Reviewer for this excellent suggestion, so we added a comprehensive section in Chapter 7 discussing the use and prospect of virus- and phage therapy. We also separated subchapters for a more organized view.
Interestingly, while there are many endeavors to modulate the microbiome to assist different treatments, especially for infectious diseases (see citations), for the specific aim to reach higher ICI efficacy or diminished irAEs, no studies have been reported so far. This makes this potential field a promising future direction for cancer Immunotherapy.
Comment 3. Text line 241: Please explain the abbreviation DMARDs
We thank the Reviewer for pointing this error out, we corrected it.
Text line 583: superscript of 13 in 3.8 x 1013
Text line 615: e>a; the addition and deletion of beneficial….
Likewise, we corrected the above errors and we thank the Reviewer for spotting these.